# Foxp1 controls brown/beige adipocyte differentiation and thermogenesis through regulating β3-AR desensitization

Pei Liu[1,7], Sixia Huang[2,7], Shifeng Ling[2], Shuqin Xu[2], Fuhua Wang[2], Wei Zhang[2], Rujiang Zhou[2], Lin He[2], Xuechun Xia[2], Zhengju Yao[2], Ying Fan[1], Niansong Wang[1], Congxia Hu[3], Xiaodong Zhao[4], Haley O. Tucker [5], Jiqiu Wang[6]* & Xizhi Guo[1,2]*

β-Adrenergic receptor (β-AR) signaling is a pathway controlling adaptive thermogenesis in brown or beige adipocytes. Here we investigate the biological roles of the transcription factor Foxp1 in brown/beige adipocyte differentiation and thermogenesis. Adipose-specific deletion of *Foxp1* leads to an increase of brown adipose activity and browning program of white adipose tissues. The *Foxp1*-deficient mice show an augmented energy expenditure and are protected from diet-induced obesity and insulin resistance. Consistently, overexpression of *Foxp1* in adipocytes impairs adaptive thermogenesis and promotes diet-induced obesity. A robust change in abundance of the β3-adrenergic receptor (β3-AR) is observed in brown/beige adipocytes from both lines of mice. Molecularly, Foxp1 directly represses β3-AR transcription and regulates its desensitization behavior. Taken together, our findings reveal Foxp1 as a master transcriptional repressor of brown/beige adipocyte differentiation and thermogenesis, and provide an important clue for its targeting and treatment of obesity.

[1] Department of Nephrology, Shanghai Jiao Tong University Affiliated Sixth People's Hospital, Shanghai 200233, China. [2] Bio-X Institutes, Key Laboratory for the Genetics of Developmental and Neuropsychiatric Disorders, Ministry of Education, Shanghai Jiao Tong University, Shanghai 200240, China. [3] School of Biomedical Engineering, Shanghai Jiao Tong University, Shanghai 200240, China. [4] Shanghai Center for Systems Biomedicine, Shanghai Jiao Tong University, Shanghai 200240, China. [5] Institute for Cellular and Molecular Biology, University of Texas at Austin, Austin, TX 78712, USA. [6] Department of Endocrinology and Metabolism, Ruijin Hospital, Shanghai Jiao Tong University School of Medicine, Shanghai 200025, China. [7] These authors contributed equally: Pei Liu, Sixia Huang. *email: wangjq@shsmu.edu.cn; xzguo2005@sjtu.edu.cn

Energy homeostasis in mammals mainly is controlled by energy intake and expenditure. There are three types of adipose tissues: white adipose tissues (WAT), brown adipose tissues (BAT), and brown-like (beige) adipose tissue. Each are composed of morphologically and anatomically distinct adipocyte populations[1,2]. WAT acts to store energy in the form of triglycerides, whereas BAT and beige adipose tissues dissipate heat against cold or excessive diet[3]. Brown adipocytes uniquely express uncoupling protein-1 (UCP1), which is the functional marker of brown adipocytes and fuels oxidation and heat generation in the inner membrane of mitochondria. Brown adipocytes originate from Myf5[+] mesodermal progenitor cells, common progenitors of skeletal muscle[4]. Beige and brown adipocytes share similar morphological features as well as expression of UCP1. Both beige and brown adipocytes can be induced within WAT depots following cold exposure or β-adrenergic stimulation[5–8]. These processes are termed browning of white adipocytes. In the context of the global obesity epidemic, mechanisms that activate brown or beige adipocytes may have significant clinical implications for metabolic disorders[9].

Activated brown or beige adipocytes are induced to dissipate heat as adaptive thermogenesis under sympathetic nervous system (SNS) control[10]. Upon cold exposure, SNS releases adrenergic signals, such as noradrenaline or catecholamines, which activate the downstream pathway through binding adrenergic receptors in adipose tissues. A variable portfolio of α/β adrenergic receptors is then assembled within various mammalian tissues. Compared to the β1/2 adrenergic receptors (β1/2-AR), the β3 adrenergic receptor (β3-AR) is relatively enriched in adipose tissues required for cold or diet-induced thermogenesis[11,12]. Adrenergic signaling triggers energy expenditure chronically via BAT thermogenesis[13,14], and acutely via WAT fueling[15,16]. β3-AR is primarily responsible for beige adipocyte induction in pre-existing white adipocytes under adrenergic agonists stimulation, whereas β1-AR mediates beige adipocyte generation after cold exposure[17]. The role of β3-AR agonists in beige induction and activation has also been demonstrated in human adipose tissues[18]. Consequently, β3-AR agonists can activate both typical brown and beige adipocytes through SNS, and thus, provide promising opportunities for anti-obesity treatments[19].

Both the abundance and activity of β3-AR in adipocytes are tightly regulated. For instance, the subjects carrying a Trp64Arg mutation within the β3-AR gene may be more susceptible to obesity[20–22]. Consistent with this finding, mice devoid of the β3-AR gene are prone to deposit more fat than control mice[23]. In ob/ob mice, β3-AR expression is dramatically impaired in adipocytes[24]. In fact, β3-AR has a unique expression dynamic in adipocytes termed desensitization. That is, β3-AR displays a short-term decline in mRNA abundance upon exposure to β3-AR agonists[25–28]. This is distinct from the typical β2-AR desensitization pathway observed in cardiomyocytes[29], which is cycled with β2-AR protein between cell membrane and endosome through β-Arrestin protein. The biological significance of β3-AR desensitization still is not fully recognized, and it remains unclear how β3-AR is transcriptionally regulated. Yet, these questions are absolutely critical in evaluating the role of β3-AR in obesity treatment.

Brown/beige adipocyte differentiation and activation is controlled by sequential actions of transcription factors, including Ebf2, Prdm16, C/ebpβ, PGC-1α and PPARγ[30–33]. The Prdm16-C/ebpβ complex functions as a switch to determine the thermogenic program of brown/beige adipocytes[34,35]. On the other hand, Twist1 and Rip140 act to arrest BAT thermogenesis by repressing PGC-1α activity[36,37]. Foxhead P1 (Foxp1) typically acts as a transcriptional repressor in a variety of developmental pathways, including cardiomyocyte proliferation[38,39], lung development[40,41], lymphocyte differentiation[42,43], glucose homeostasis[44], endochondral ossification[45], and neuronal morphogenesis[46–48]. A recent study from our group further reveals an important role for Foxp1 in mesenchymal stem cell senescence[49]. In this study, we identify Foxp1 as a crucial component of the thermogenic program, which arrests brown/beige differentiation and thermogenesis through regulation of β3-AR transcription in adipocytes.

## Results

**Foxp1 expression is sensitive to adrenergic stimuli.** To examine the expression pattern of Foxp1 in adipose tissues, two representative subpopulations of adipocytes, interscapular BAT and subcutaneous WAT were investigated by immunofluorescence analyses. Foxp1 expression was strongly detected within brown and white adipocytes from 4-week-old mice (Fig. 1a). Of the four isoforms (A–D) that are typically observed in a variety of mouse tissues[50], we detected primarily isoforms B and D in BAT, and isoforms A and B in WAT via western blotting analyses (Fig. 1b). In pheochromocytoma (PHEO) patients, beige adipocytes were induced inside omental WAT as a result of adrenergic stress under extremely excessive catecholamine expression[51,52]. In clinical samples from PHEO patients, we detected enrichment of FOXP1 expression in beige adipocytes in the vicinity of the vasculature within omental WAT (Fig. 1c and Supplementary Fig. 1a).

Then stromal vascular fraction (SVF) cells isolated from BAT of wild type mice were induced to brown adipocyte differentiation in vitro. After one day of induction, the elevated Foxp1 expression began to decline, whereas the expression of brown adipocyte-related genes (Ucp1, C/ebpα and PPARγ) was progressively upregulated (Fig. 1d). Similarly, qPCR analyses confirmed the downregulation of Foxp1 during the 8-hour course of white adipogenic induction in 3T3-L1 cells (Supplementary Fig. 1b), which was consistent with previous findings[53]. These observations indicate that Foxp1 is extensively expressed in adipocytes, and its expression shows a transit peak at the very earliest stage of adipocyte differentiation.

Next, we examined the dynamics of Foxp1 expression following stimulation of adrenergic signaling. When mice were challenged by cold exposure (4 °C) overnight, Foxp1 expression in BAT was upregulated (Fig. 1e, f). At the cellular level, when brown adipocytes derived from SVF of mice or humans were exposed to CL-316,243 (0.1 μM) for up to 8 h, Foxp1 expression progressively inclined, whereas β3-AR expression inversely declined (Fig. 1g, h). Consistently, the expression of β3-AR behaved as a typical desensitization process at the transcriptional level in responsive to adrenergic signaling in vivo and in vitro (Fig. 1e, g, h). These observations suggest that Foxp1 expression in adipose adipocytes is dynamic, and can be induced by adrenergic signaling with an inverse expression pattern of β3-AR.

Previous studies have demonstrated that MAPK kinases, including Erk1/2 and p38, could mediate adrenergic signaling in adipocytes[15,54]. To test if either of these kinases mediates induction of Foxp1 by the β3-AR agonist, we first evaluated the dynamics of Foxp1 expression in an 8-h course of adipogenic cultures of 3T3-L1 cells. Briefly, the cells were stimulated with CL-316,243 alone or in combination with SB202190 (p38 kinase inhibitor), FR180204 or SCH772984 (Erk1/2 inhibitors). As shown in Fig. 1i, the induction of Foxp1 transcripts by the β3-AR agonist was markedly arrested by FR180204 and SCH772984, whereas it was exaggerated by SB202190. This suggested that Erk1/2 is necessary for Foxp1 induction by adrenergic stimulation. These results were further validated by western blotting following exposure of brown adipocytes to CL-316,243 and SCH772984 (Fig. 1j). Together our findings indicate that Foxp1

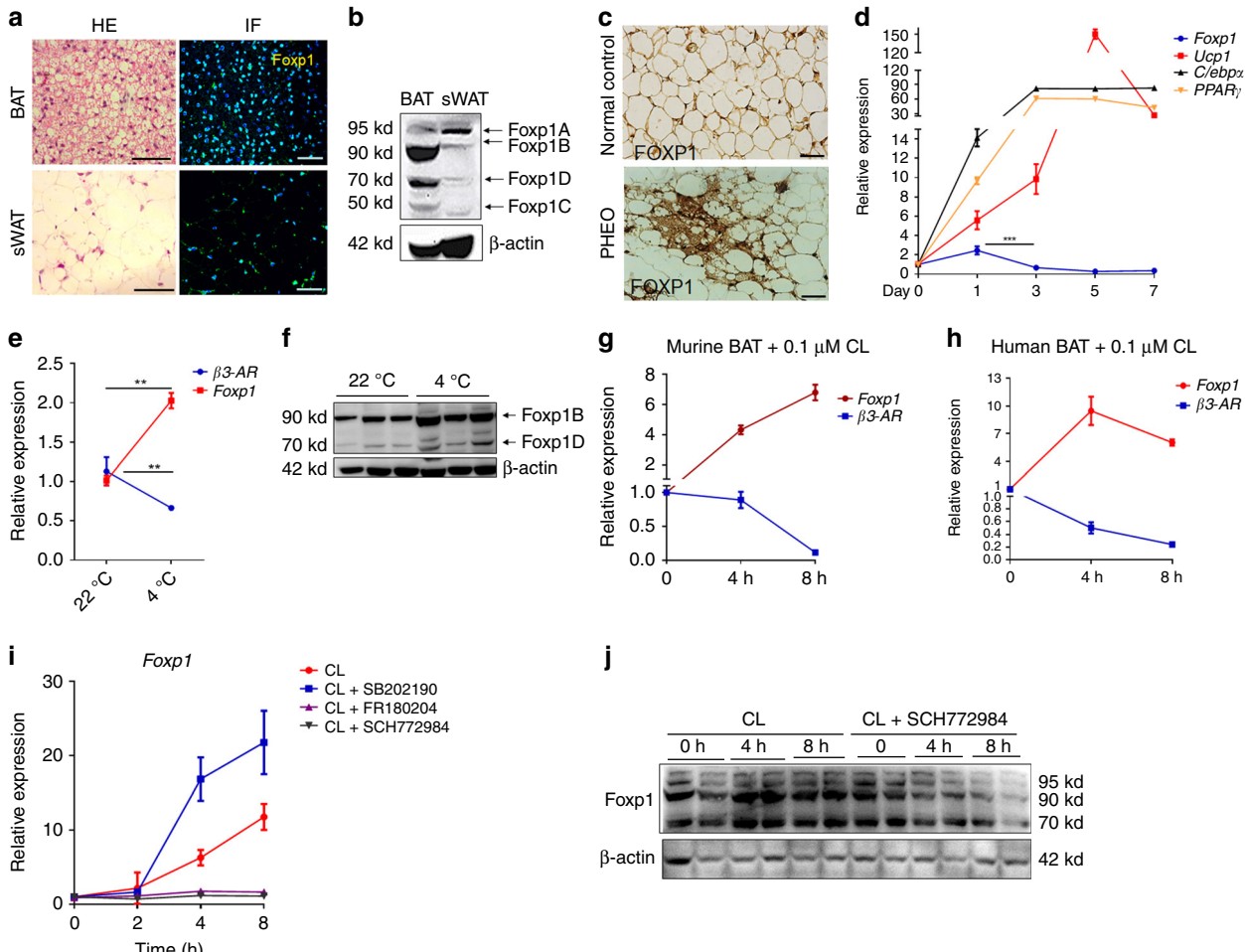

**Fig. 1** Foxp1 expression in adipocytes is induced by adrenergic stimuli. **a** H&E staining and immunofluorescence (IF) analysis for the Foxp1 expression in BAT and sWAT from wild type mice at age of 4 weeks. DAPI, blue staining for nucleus; green color for Foxp1 expression. Bar, 50 μm. **b** Western blotting showed the four isoforms (A, B, D, C) of Foxp1 protein in BAT and sWAT from wild type mice at age of 2 months. **c** IHC analysis of FOXP1 expression in biopsies from PHEO patients and normal controls. Bar, 10 μm. **d** qPCR analysis of expression of *Foxp1* and brown adipocyte markers (*C/ebpα*, *Pparγ*, and *Ucp1*) during the time course of brown adipocyte differentiation from SVFs. **e** qPCR analysis of *Foxp1* and *β3-AR* expression in BAT in mice with overnight 4 °C cold exposure. $n = 3$ biologically independent samples. **f** Western blotting of Foxp1 in BAT from mice above (**e**). **g, h** qPCR analysis of *Foxp1* and *β3-AR* expression in brown adipocytes differentiated from murine (**g**) and human SVF (**h**) during an 8-hour CL-316,243 (0.1 μM) treatment as indicated. $n = 3$ biologically independent experiments. **i** *Foxp1* expression profile in adipocytes derived from 3T3-L1 cells during an 8-h time course, stimulated by CL-316,243 (0.5 μM) with or without SB202190 (p38 kinase inhibitor, 10 μM), FR180204 (Erk1/2 inhibitors, 1 μM) and SCH772984 (Erk1/2 inhibitors, 10 μM), respectively. $n = 3$ biologically independent experiments. **j** Western blotting for Foxp1 in brown adipocytes derived from SVF, which were stimulated by CL-316,243 (0.1 μM) with or without SCH772984 (10 μM) for 8 h. *$P < 0.05$; **$P < 0.01$; ***$P < 0.001$; error bar, mean ± SEM

expression in adipocytes is induced by a β3-AR/Erk1/2 signal cascades.

***Foxp1* deletion potentiates brown adipocyte differentiation.** As a next step in investigating the function of *Foxp*1 in brown adipocyte differentiation, we generated $Foxp1_{Myf5}^{\Delta/\Delta}$ mice by crossing *Foxp1*^fl/fl with *Myf5-Cre* to facilitate deletion of *Foxp1* specifically within BAT progenitors[4] (Fig. 2c and Supplementary Fig. 1c). At the early postnatal stage, The BAT appeared normal in morphology and histology in $Foxp1_{Myf5}^{\Delta/\Delta}$ mice as compared to controls (Fig. 2a and upper panel in Fig. 2b). However, immunohistochemistry (IHC) analysis revealed a relative increase of UCP1 expression in the BAT of $Foxp1_{Myf5}^{\Delta/\Delta}$ mice (Fig. 2b, lower panel), which was validated by western blotting (Supplementary Fig. 1c). In addition, qPCR detected elevated expression of BAT-related (*Ucp1*, *Dio2*, *Prdm16*, *Tbx1*, *PPARγ*) and mitochondrial (*Cox7a1*, *Cox8b*, *Cox5b*, *Cox2*, *Cpt2*) genes in *Foxp1-*$_{Myf5}^{\Delta/\Delta}$ BAT (Fig. 2d). These observations indicate that *Foxp1*

deletion accelerates brown adipocyte differentiation from progenitor cells.

***Foxp1* deficiency promotes adipose tissue browning.** To observe any potential influence of Foxp1 deficiency on beige adipocyte differentiation in vivo, we eliminated *Foxp1* in adipocytes with *Adiponectin-Cre*[55], which thereafter designated as $Foxp1_{Ad}^{\Delta/\Delta}$. Compared to control littermates, $Foxp1_{Ad}^{\Delta/\Delta}$ knockout mice appeared normal size (Fig. 2e), but was slightly smaller in BAT and sWAT depots (Fig. 2f). H&E staining detected a reduction in the size of white adipocytes in $Foxp1_{Ad}^{\Delta/\Delta}$ sWAT in relative to controls (Fig. 2g). Typical browning characteristics of adipose tissues were detected by anti-UCP1 IHC of sWAT in $Foxp1_{Ad}^{\Delta/\Delta}$ mice exposed to 6-h 4 °C challenge (Fig. 2h). The activation of brown or beige adipocytes were observed by the elevated expression of a broad panel of thermogenic or BAT-selective marker genes (*Ucp1*, *PGC-1α*, *PGC-1β*, *Dio2*, *Cidea*, *Otop1*, *Prdm16*, *PPARα*, *Cox7a1*, *Eva1*, *Cox4il*, *Cox8b*) (Fig. 2j, k),

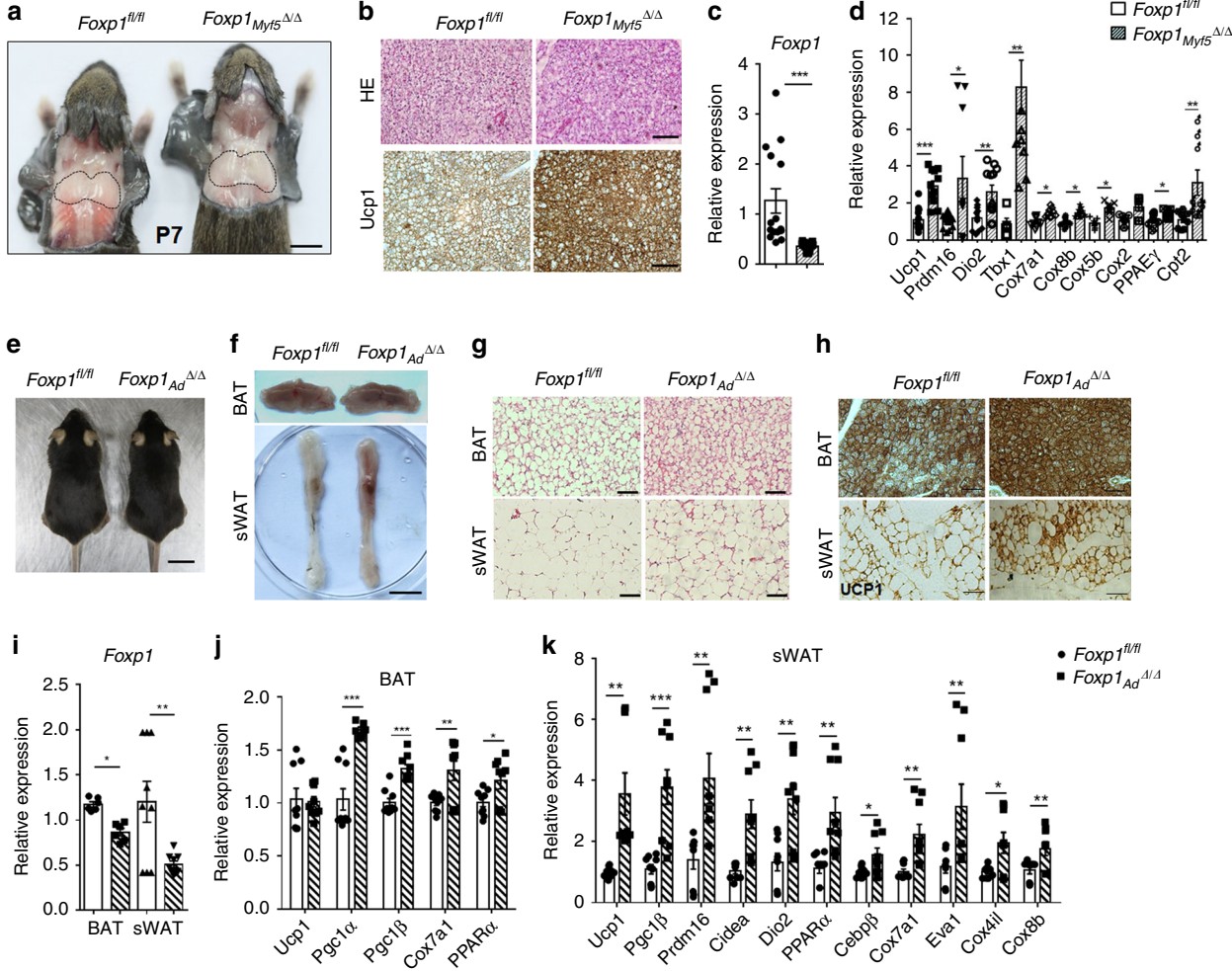

**Fig. 2** *Foxp1* deficiency promotes brown and beige adipocyte activation. **a** Representative dorsal view of BAT depot in *Foxp1^{fl/fl}* and *Foxp1_{Myf5}^{Δ/Δ}* mice at postnatal day 7. Bar, 1 cm. **b** HE (upper panel) and IHC staining (lower panel) with UCP1 antibody for BAT in (**a**). Bar, 10 μm. **c, d** qPCR analysis of *Foxp1* and BAT-selective markers for BAT depot. *n* = 5 biologically independent samples. **e** Representative view of 3-month-old *Foxp1^{fl/fl}* and *Foxp1_{Ad}^{Δ/Δ}* mice. Bar, 2 cm. **f** Representative view of BAT (upper panel) and sWAT depot (lower panel) in (**e**). **g** HE staining for BAT and sWAT in mutant mice. Bar, 50 μm. **h** IHC analysis with UCP1 antibody for BAT and sWAT upon 6-h cold exposure. Bar, 50 μm. **i–k** qPCR for expression of *Foxp1* and BAT-selective genes in BAT and sWAT from *Foxp1^{fl/fl}* and *Foxp1_{Ad}^{Δ/Δ}* mutant mice. *n* = 4 biologically independent mice/each group. *$P < 0.05$; **$P < 0.01$; ***$P < 0.001$; error bar, mean ± SEM

whereas the *Foxp1* expression was significantly decreased in mutant BAT and sWAT (Fig. 2i).

We also ablated *Foxp1* in adipocytes by *aP2-Cre*, which was also extensively utilized for adipose-specific knockout models[56]. Foxp1 was efficiently reduced at the mRNA and protein levels in BAT and sWAT from *Foxp1_{aP2}^{Δ/Δ}* mice (Supplementary Fig. 1d, e). Similar to *Foxp1_{Ad}^{Δ/Δ}*, *Foxp1_{aP2}^{Δ/Δ}* mice were also smaller size in adipose depot (Supplementary Fig. 2a, b). H&E staining and transmission electron microscopic (TEM) analysis revealed smaller lipid droplets as well as relative enrichment in mitochondria within brown and beige adipocytes of the mutant mice (Supplementary Fig. 2c, d), as evidenced by increased expression of mitochondrial specific gene, *Co.1* (Supplementary Fig. 2e). In addition, adipose tissue browning in *Foxp1_{aP2}^{Δ/Δ}* mice were observed by anti-UCP1 immunofluorescence (Supplementary Fig. 2f), elevated expression of BAT-selective genes (Supplementary Fig. 2g), as well as western blotting with anti-UCP1 and anti- PGC-1α (Supplementary Fig. 2h). *Foxp1*-deficient adipose tissues also displayed elevated expression of lipolytic genes, including *Hsl*, *Atgl*, *Acsl*, and *Acox1* (Supplementary Fig. 2i), and augmented p38 and HSL phosphorylation

(Supplementary Fig. 2j)—key mediators or targets of β3-AR signaling for lipolysis[13]. Taken with our observations in *Foxp1_{Myf5}^{Δ/Δ}*, *Foxp1_{Ad}^{Δ/Δ}* and *Foxp1_{aP2}^{Δ/Δ}* mice, these findings suggest that Foxp1 suppresses both brown and beige adipocyte differentiation and thermogenesis.

To test for a potential cell-autonomous effect of Foxp1 loss on brown/beige differentiation, stromal vascular fraction (SVF) cells were isolated from BAT or sWAT depots of *Foxp1_{aP2}^{Δ/Δ}* and control mice and then induced for brown adipocyte differentiation in vitro. We observed advanced brown/beige adipocyte differentiation from the *Foxp1*-deficient SVF progenitors (Supplementary Fig. 3a, e). Further, expression of a set of BAT-selective (*Ucp1*, *PPARα*, *PGC-1α*, *PGC-1β*, *Otop1*, *Dio2*, *Tbx1*) and mitochondrial (*Cox7a1*, *CytoC*, *Cpt2*, *Cox2*, *Cox5b*, *Cox8b*) transcripts were significantly upregulated in cultured brown and beige adipocytes (Supplementary Fig. 3b–h). We next analyzed oxygen consumption rates (OCR) of SVF-derived adipocytes. Brown adipocytes from *Foxp1_{aP2}^{Δ/Δ}* mutant mice exhibited higher total and uncoupled OCR (Supplementary Fig. 3i). Similarly, OCR was each significantly elevated in the white adipocytes from *Foxp1_{aP2}^{Δ/Δ}* relative to controls (Supplementary

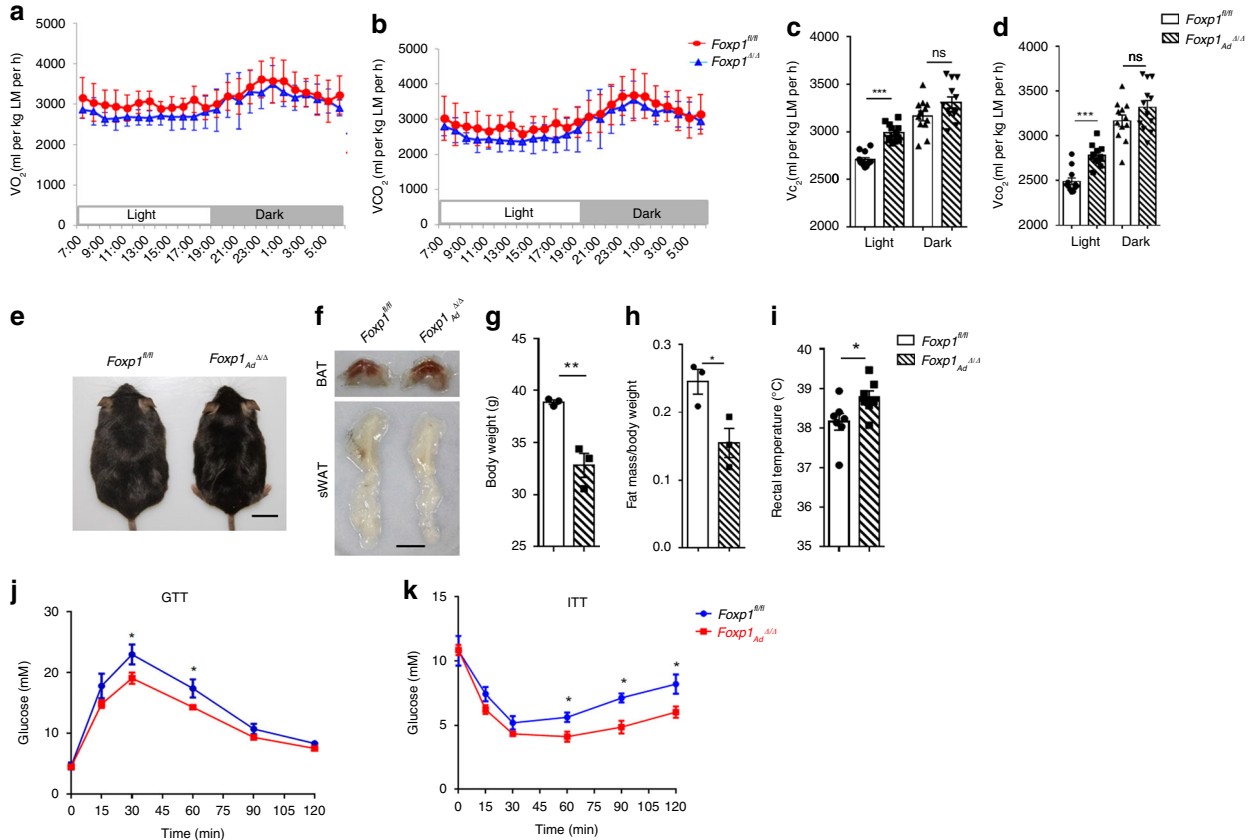

**Fig. 3** *Foxp1* deletion elevates energy expenditure and resists to HFD-induced obesity. **a**, **b** $VO_2$ and $VCO_2$ of $Foxp1^{fl/fl}$ and $Foxp1_{Ad}{}^{\Delta/\Delta}$ mice in metabolic cages at age of 3 months old. $n = 7$ biologically independent mice/each group. **c**, **d** Quantification of $O_2$ and $CO_2$ consumption in light and dark. **e** Dorsal view of $Foxp1^{fl/fl}$ and $Foxp1_{Ad}{}^{\Delta/\Delta}$ mice after 8-week feeding with HFD at age of 6 months old. Bar, 2 cm. **f** Representative depot of BAT and sWAT in HFD-fed mice (**e**). Bar, 1 cm. **g** Body weight of HFD-fed mice. **h** Relative adiposity of HFD-fed mice. **i** Statistics of rectal temperature of HFD-fed mice. **j**, **k** GTT and ITT of HFD-fed mice. $*P < 0.05$; $**P < 0.01$; $***P < 0.001$; $n = 7$ biologically independent mice/each group; error bar, mean ± SEM

Fig. 3j). These lines of evidence indicate that Foxp1 controls brown/beige adipocyte differentiation and thermogenesis in a cell-autonomous manner.

**$Foxp1$ deletion augments thermogenesis and energy expenditure.** As BAT activation and sWAT browning are features of thermogenesis and energy expenditure, we next examined energy metabolism in mutant mice. $Foxp1_{Ad}{}^{\Delta/\Delta}$ and $Foxp1_{aP2}{}^{\Delta/\Delta}$ mice exhibited higher rates of oxygen consumption and carbon dioxide production, expended more energy (Fig. 3a–d and Supplementary Fig. 4a–d). And more, infrared imaging revealed that $Foxp1_{aP2}{}^{\Delta/\Delta}$ mice displayed higher skin temperatures as compared to $Foxp1^{fl/fl}$ controls after 1-hour exposure to 4 °C (Supplementary Fig. 4e, f). These observations are consistent with that $Foxp1$ deficiency increases energy expenditure and thermogenesis in vivo.

To examine the long-term effect of $Foxp1$ loss on energy balance, mutant mice were subjected to a high-fat diet (HFD). $Foxp1_{Ad}{}^{\Delta/\Delta}$ mice appeared leaner in body, with smaller in adipose depots (Fig. 3e, f), and gained less body weight than littermates after 8-week HFD feeding starting at age of 6 months old (Fig. 3g). The reduction in weight primarily resulted from a decrease in adiposity (Fig. 3h). Meanwhile, $Foxp1_{Ad}{}^{\Delta/\Delta}$ mice appeared relatively high rectal temperature as compared to controls (Fig. 3i). $Foxp1_{Ad}{}^{\Delta/\Delta}$ mutant mice also retained better glucose tolerance and higher insulin sensitivity following HFD feeding, as evidenced by glucose tolerance test (GTT) and insulin tolerance test (ITT) scores (Fig. 3j, k). $Foxp1_{aP2}{}^{\Delta/\Delta}$ knockout mice

displayed similar phenotypes with $Foxp1_{Ad}{}^{\Delta/\Delta}$ mice, including resistance to HFD-induced obesity as well as improved glucose metabolism and insulin sensitivity after HFD feeding (Supplementary Fig. 4g–l). Our findings suggest that BAT activation and WAT browning in $Foxp1$-deficient mice results in marked improvement of glucose metabolism and protects mice from HFD-induced obesity (potentially as a result of elevated thermogenesis).

**Overexpression of $Foxp1$ represses adaptive thermogenesis.** To further examine the effect of $Foxp1$ on energy expenditure, we generated $aP2$-$Foxp1$ transgenic mice in which $Foxp1$ cDNA overexpression was driven by a 5.4 kb $Fabp4$ ($aP2$) gene promoter/enhancer cassette. Overexpression of $Foxp1$ was validated by qPCR and western blotting of four independent founders (Supplementary Fig. 5a, b). We then employed Thermo Mouse, an $Ucp1$-$Luciferase$ ($Ucp1$-$Luc$) reporter mouse[57], as a tool to detect adaptive thermogenesis in brown and beige adipose tissues under cold exposure or adrenergic stimulation. Two representative strains of $aP2$-$Foxp1;Ucp1$-$Luc$ transgenic mice displayed lower luciferase activities in BAT and sWAT compared to $Ucp1$-$Luc$ controls when challenged with 4 °C cold exposure for 6 h (Fig. 4a). Transgenic BAT depots appeared less brown in color and larger in cellular size as compared to wild type controls (Fig. 4b, c). We also observed that our transgenic mice displayed impaired thermogenesis under cold exposure, as evidenced by decreased UCP1

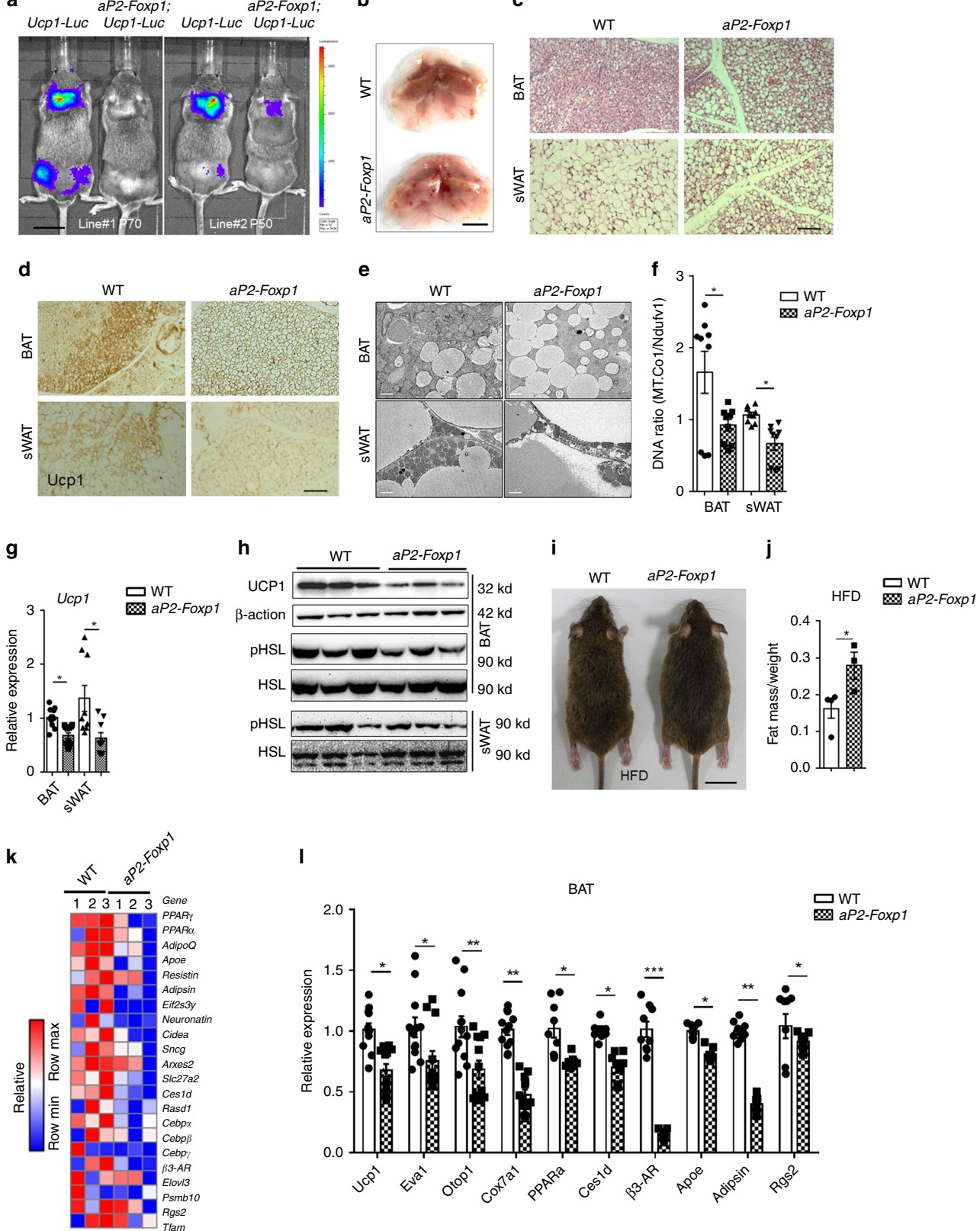

induction within BAT and sWAT (Fig. 4d, g, h) and less abundance of mitochondria (Fig. 4e, f). These observations suggest that overexpression of *Foxp1* disrupts the thermogenic program in adipose tissues.

Finally, our transgenic mice suffered perturbed lipolysis, as shown by their decreased phosphorylation of HSL (Fig. 4h)—a phenotype opposite to $Foxp1_{aP2}{}^{\Delta/\Delta}$ mutant mice (Supplementary Fig. 2j). As a consequence, the transgenic mice appeared to be sensitive to HFD-induced obesity (Fig. 4i, j). Collectively, these phenotypes were similar to those of previously reported mutant mice, such as BAT-specific *Prdm16* knockout and global *β3-AR* knockout mice[23,58]. Taken together, the potency of adaptive

**Fig. 4** Overexpression of *Foxp1* represses adaptive thermogenesis and is sensitive to HFD-induce obesity. **a** Representative bioluminescence imaging of two stains of *aP2-Foxp1;Ucp1-Luc* transgenic mice after 6-hour cold exposure. $n = 3$ biologically independent mice/each group. Bar, 2 cm. **b** Representative picture of BAT depot in *aP2-Foxp1* transgenic mice. Bar, 0.5 cm. **c** H&E staining for adipose sections of BAT and sWAT from transgenic mice in (**b**). Bar, 10 μm. **d** IHC with UCP1 antibody for adipose sections from transgenic mice after 6-h cold exposure. Bar, 10 μm. **e** Intracellular structure of adipocytes in BAT and sWAT from *aP2-Foxp1* mice, as showed by transmission electronic microscope (TEM). Bar, 2 μm. **f** Assessment of mitochondria DNA abundance by qPCR for the BAT and sWAT from transgenic mice. $n = 4$ biologically independent mice/each group. **g** qPCR analysis for *Ucp1* expression in adipose tissues of transgenic mice. $n = 4$ biologically independent samples. **h** Western blotting for UCP1, HSL, and phosphorylated HSL in BAT and sWAT. **i** Representative dorsal view of *aP2-Foxp1* transgenic mice subjected to 8-week HFD feeding since at age of 3 months. Bar, 2 cm. **j** Relative adiposity in transgenic mice (**i**). $n = 4$ biologically independent mice/each group. **k** The heat map of relative expression of adipogenic genes in three strains of *aP2-Foxp1* transgenic mice at age of 3 weeks, as evaluated by RNA-seq. **l** qPCR analysis validated the BAT-selective genes expression in BAT from (**k**). $n = 3$ biologically independent mice/each group; *$P < 0.05$; **$P < 0.01$; ***$P < 0.001$; error bar, mean ± SEM

thermogenesis is impaired in *aP2-Foxp1* transgenic mice in the opposite direction as $Foxp1_{aP2}^{\Delta/\Delta}$ knockout mice.

To address molecular mechanisms utilized by Foxp1 in regulating the brown/beige program, we first compared gene expression profiles of BAT from 3-week-old wild type and *aP2-Foxp1* mice by RNA-seq. As shown in the heat map of Fig. 4k, the expression levels of a set of BAT-selective genes (*Ucp1*, *Eva1*, *PPARα*, *Otop1*, *Cox7a1*, *Ces1d*) was down-regulated within the BAT of *aP2-Foxp1* mice relative to controls. Next, we employed qPCR to validate the expression profiles of several marker genes in the BAT of transgenic mice. A significant decrease of β3-AR expression was observed among a broad panel of thermogenic genes (Fig. 4l). The changes in those gene expression profiles may underlie the defective thermogenesis in transgenic mice.

**Foxp1 directly represses β3-AR transcription in adipocytes**. β3-AR is selectively expressed in the adipose tissues which mediate adaptive thermogenesis in responsive to adrenergic stimuli[11,12]. qPCR and western blotting analyses revealed a decrease in β3-AR expression within the BAT and sWAT of *aP2-Foxp1* transgenic mice (Fig. 5a, b), and reciprocally, an increase of β3-AR expression in $Foxp1_{aP2}^{\Delta/\Delta}$ knockout mice (Fig. 5c, d). These data suggested that alteration of β3-AR expression may partially account for the disturbance of thermogenesis in both *Foxp1* knockout and transgenic mice. Of note, a consensus Foxp1 binding site (TTATTTAT) was detected at −2251 bp upstream of the *β3-AR* promoter (Fig. 5f). This led us to conduct promoter occupancy analysis by ChIP-seq and ChIP-PCR based on SVF progenitor cells, confirming this site within the chromatin of the *β3-AR* promoter (Fig. 5e, f). In addition, the increased OCR of SVF-derived brown adipocytes in $Foxp1_{aP2}^{\Delta/\Delta}$ knockout mice (Supplementary Fig. 3i), was arrested either by exposure to the β3-AR antagonist L748337, or by knockdown with *β3-AR*-shRNA (Fig. 5g). Infrared imaging revealed a decline of dorsal skin temperature in *aP2-Foxp1* transgenic mice (Fig. 5h). Normal temperature was restored when the mutant mice were challenged by the β3-AR agonist CL-316,243 (Fig. 5i). Thus, the β3-AR agonist could rescue the thermogenic defects of the transgenic mice (Fig. 5h), whereas β3-AR antagonist could block the enhanced thermogenic phenotypes observed in the knockout mice (Fig. 5g). Furthermore, we observed that brown adipocytes derived from SVF of $Foxp1_{aP2}^{\Delta/\Delta}$ knockout and *aP2-Foxp1* transgenic mice altered the desensitization behavior of β3-AR, as evidenced by the β3-AR expression profiles during 6-hours treatment with CL-316,243. As shown in Fig. 5j, the down-regulation of *β3-AR* in brown adipocytes from $Foxp1_{aP2}^{\Delta/\Delta}$ mice declined sharply as compared to the wild type controls, whereas those of transgenic mice declined more smoothly. Together these findings suggest that Foxp1, at least in partial, controls thermogenesis through repressing of *β3-AR* transcription within brown/beige adipocytes.

**Foxp1 forms a complex with Prdm16-C/ebpβ proteins**. We previously observed that Foxp1 regulates the adipogenic potential of mesenchymal stem cells by interacting with the C/ebpβ/δ complex[49]. Both C/ebpβ and Prdm16 are critical regulators of brown adipocyte differentiation and thermogenesis[34]. Here, we observed via co-immunoprecipitation (Co-IP) assays that Foxp1 interacted with C/ebpβ and Prdm16 in SVF cells isolated from BAT (Fig. 6a). We validated these results in HEK293T cells co-transfected with Foxp1-His, Prdm16-Flag and C/ebpβ-Myc expression plasmids (Fig. 6b). In support, immunofluorescence analysis also detected co-localization of Foxp1 and Prdm16 within the nucleus of brown adipocytes (Fig. 6c). Luciferase reporter assays employing a *PPARγ* promoter-driven luciferase vector further showed that Foxp1 repressed transactivation ability of Prdm16 in 3T3-L1 cells (Fig. 6d). Similarly, assays employing a *β3-AR* promoter-driven luciferase vector revealed that Foxp1 repressed transcription of the *β3-AR* by antagonizing the trans-activation ability of the Prdm16-C/ebpβ complex (Fig.6e). Collectively, these data suggest a mechanism by which Foxp1 forms a complex with Prdm16-C/ebpβ proteins to repress *β3-AR* transcription (Fig. 6f).

**Discussion**

In mammals, brown/beige thermogenesis is triggered to dissipate heat for thermoregulation upon cold challenge, mainly through a thermogenic program mediated by β-AR signaling[59]. Thus, the activity of β-AR signaling has to be precisely controlled. In pheochromocytoma (PHEO) patients, WAT browning was detected as a result of severe adrenergic stress under continuously high levels of catecholamines, which disrupted the energy balance in adipose tissues to result in weight loss[51]. To restrain this overactivation of BAT thermogenic program, a number of tran-scriptional co-repressors has been employed to counteract the activity of PGC-1α, e.g. Rip 140 and Twist1[36,37], or to repress transcription of *Ucp1*, e.g. Rb1[60] and LXRα[61]. In this study, we examined the influence of *Foxp1* on thermogenesis in adipocytes from knockout mice employing *Myf5-Cre*, *aP2-Cre* and *Adiponectin-Cre*, as well as by transgenic mice with *Foxp1* over-expression in adipocytes. Our analyses identify Foxp1 as a repressor for brown/beige adipocyte differentiation and energy expenditure. Luciferase reporter suggested that Foxp1B, the iso-form that was expressed in BAT and sWAT, mostly accounted for the repressive function in BAT activation and thermogenesis (Supplementary Fig. 5c). We further show that Foxp1 expression could be induced by a β3-AR/cAMP/Erk1/2 cascade. As a nega-tive feedback brake, Foxp1 acts to repress thermogenesis through antagonizing the action of the Prdm16-C/ebpβ complex to repress *β3-AR* transcription (Fig. 6f). We suggest that this circuit may precisely control *β3-AR* transcription to maintain the set-point of body weight.

According to the robust phenotypes in the four categories of mutant mice, Foxp1 performs a consistent function in the

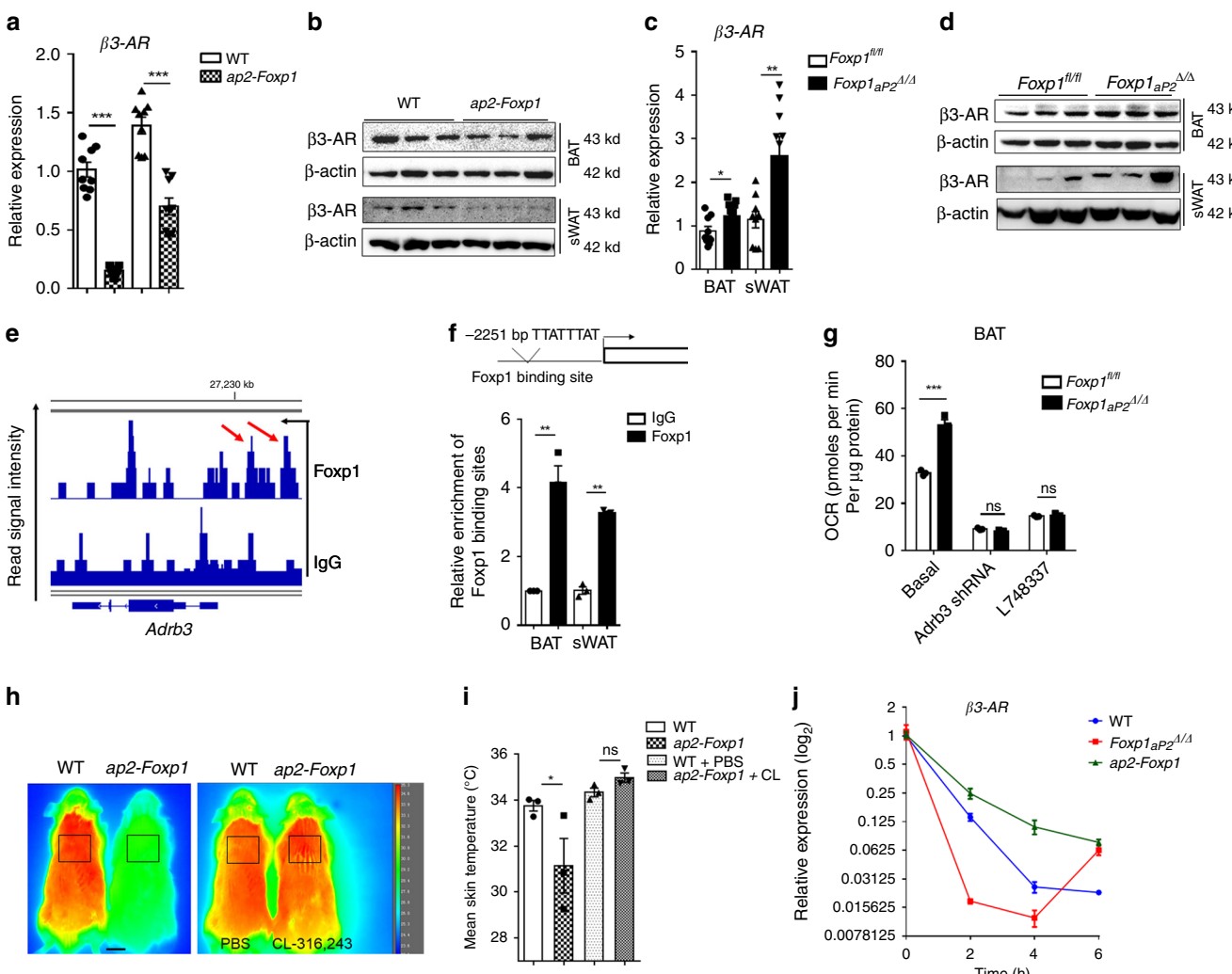

**Fig. 5** Foxp1 represses *β3-AR* transcription in adipocytes. **a, b** β3-AR expression in BAT and sWAT were assessed by qPCR and western blotting in *aP2-Foxp1* transgenic mice at 2 month old. $n = 3$ biologically independent samples/each group. **c, d** β3-AR expression in BAT and sWAT were assessed by qPCR and western blotting in *Foxp1_{aP2}^{Δ/Δ}* knockout mice at 2 months old. $n = 3$ biologically independent samples/each group. **e** ChIP-seq profile showed the Foxp1 binding sites within *Adrb3* gene promoter region. Red arrows indicated the potential Foxp1 binding sites. *Adrb3* gene was reversely transcribed. ChIP was based on SVF cells derived from BAT. **f** Upper panel showed the location of Foxp1 binding site at −2251 bp upstream of *β3-AR* gene promoter with a schematic drawing. ChIP-PCR confirmed the relative enrichment of Foxp1 binding sites (lower panel). $n = 3$ biologically independent samples/each group. **g** Basic OCR of SVF-derived brown adipocytes from *Foxp1_{aP2}^{Δ/Δ}* mice, which were transfected *β3-AR*-shRNA lentivirus or administrated with β3-AR inhibitor L748337 (10 μM) for 1 h. $n = 3$ biologically independent experiments. **h** Representative infrared imaging of *aP2-Foxp1* transgenic mice with or without 6-hour CL-316,243 (10 μM) exposure. Bar, 1 cm. **i** Quantification of average skin temperature in boxed regions in (**h**). $n = 4$ biologically independent mice/each group. Samples were isolated from mice at age of 2 months. **j** CL-stimulated β3-AR desensitization in SVF-derived brown adipocytes from *aP2-Foxp1* transgenic mice and *Foxp1_{aP2}^{Δ/Δ}* knockout mice. β3-AR expression was assessed by qPCR during the 6-h time course of CL-316243 (0.1 μM) treatment. $n = 3$ biologically independent samples/each group. *$P < 0.05$; **$P < 0.01$; ***$P < 0.001$; error bar, mean ± SEM

multiple steps of adipocyte differentiation and metabolism. Depletion of *Foxp1* via *Myf5-Cre* or *Ad-Cre* in adipocytes potentiated brown or beige adipocyte differentiation under mildly ambient temperature (Fig. 2). We previously observed that Foxp1 suppresses adipogenic commitment of mesenchymal stem cells by repressing *PPARγ* transcription[49]. Here we show that Foxp1 similarly regulates early commitment/differentiation of brown adipocytes from progenitor cells (Fig. 2a–d), through repressing the transactivation ability of Prdm16-C/ebpβ, two key regulators of brown or beige adipocyte differentiation[4,62]. Similar defects in thermogenesis and energy expenditure were observed in both *Foxp1_{aP2}^{Δ/Δ}* and *Foxp1_{Ad}^{Δ/Δ}* mice. Of note, we noticed that defects in adaptive thermogenesis in *Foxp1_{aP2}^{Δ/Δ}* mice were much more penetrant than that in *Foxp1_{Ad}^{Δ/Δ}* mice at comparable ages. White adipose tissue browning was only evident in

*Foxp1_{Ad}^{Δ/Δ}* mice under several-hours cold challenge (Fig. 2h). Elevated energy expenditure also was considerably more pronounced in *Foxp1_{aP2}^{Δ/Δ}* mice than in *Foxp1_{Ad}^{Δ/Δ}* mice (Fig. 3 and Supplementary Fig. 4). This discrepancy in thermogenic defects may result from differences in the timing and specificity of Cre activities of the two strains; i.e., the latter occurred earlier than the former as previously suggested[63]. Interestingly, Foxp1 regulates liver glycogenesis[44], thus the impact of Foxp1 on glucose homeostasis, which remains to be tested in adipocytes, may also underpin its regulation of energy metabolism described here.

Distinct subtypes of β-ARs have been shown to possess different properties of desensitization within different tissues. For example, the agonist-stimulated desensitization of β1 or β2-AR is fulfilled by β-Arrestin-dependent sequestration through receptor

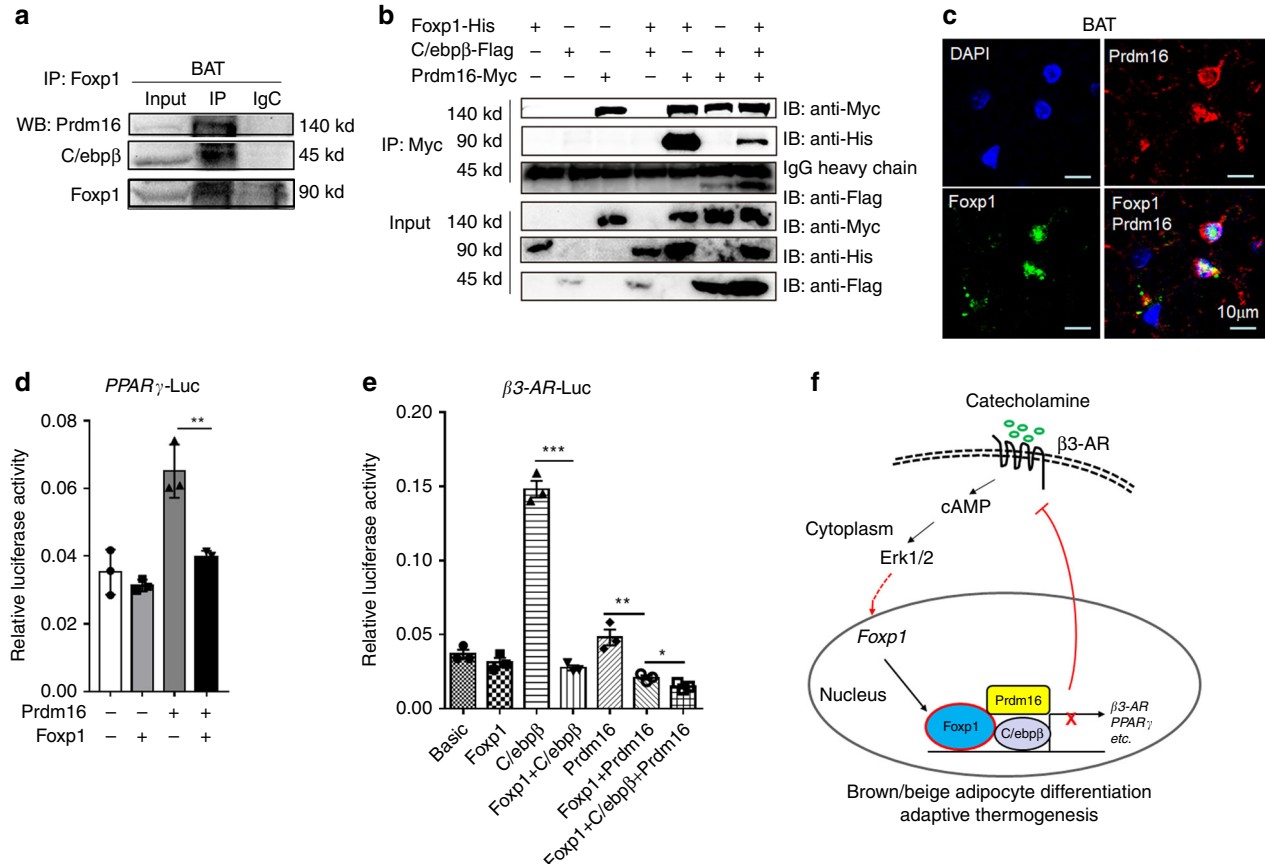

**Fig. 6** Foxp1 physically interacts with Prdm16-C/ebpβ complex in brown adipocytes. **a**, **b** Co-immunoprecipitation validated the physical interaction between Foxp1, C/ebpβ and Prdm16 proteins in cell lysates from 293T cell lines transfected with indicated plasmids (**a**), or brown adipocytes induced from stromal vascular fractions (**b**). **c** Immunofluorescence analysis showed the co-localization of Foxp1 and Prdm16 in the nucleus of BAT from wild type mice at age of 1 month old. Bar, 10 μm. **d** Luciferase reporter assay showed the transactivation of PPARγ-Luc by Foxp1 and Prdm16 protein in 3T3-L1 cell lines. n = 3 biologically independent samples/each group. **e** Luciferase reporter assay showed the transactivation of β3-AR-Luc by Foxp1, C/ebpβ and Prdm16 protein in 293T cell lines. n = 3 biologically independent experiments; *P < 0.05, **P < 0.01, ***P < 0.001; error bar, mean ± SEM. **f** Diagram depicting the mechanisms that Foxp1 represses brown/beige adipocyte differentiation and adaptive thermogenesis. Expression of Foxp1 is induced by β3-AR/cAMP/Erk1/2 cascades. Conversely, Foxp1 suppresses β3-AR transcription by counteracting the activity of Prdm16-C/ebpβ complex

phosphorylation by GRK[29,64]. In contrast, β3-AR displays resistance to typical desensitization due to a lack of putative GRK phosphorylation sites[65,66]. Instead, β3-AR may adopt a desensitization strategy primarily based on decreasing transcriptional expression in adipocytes[25–28]. Our findings may provide a clue toward understanding the mechanisms underlying the unique strategy of β3-AR desensitization. Adrenergic stimuli swiftly induced Foxp1 mRNA and protein level in adipocytes, which conversely decreased β3-AR mRNA levels. As showed in Fig. 6j, alterations of Foxp1 abundance profoundly affected the sensitivity of β3-AR desensitization. Of note, it has to be kept in mind that the β3-AR pathway only partially account for the function of Foxp1 in BAT activation and thermogenesis, given that Foxp1-deficient cells exhibit higher cell-autonomous ability to differentiate into brown adipocytes or beige adipocytes in cultured conditions without β3-AR agonist stimulation. Recently, a new type of glycolytic beige adipocytes, which are β-AR independent and derived from MyoD+ progenitor cells, is characterized in WAT upon prolonged cold exposure[67]. Surprisingly, we also observed upregulation of glycolytic beige-related marker genes (MyoD, Myh1, GABPα) in Foxp1_aP2^{Δ/Δ} WAT (Supplementary Fig. 6a). These findings suggest that Foxp1 may also regulate glycolytic beige adipocyte activation through a β-AR independent manner. ChIP-seq analysis also suggested that Foxp1 may exert extensive influence on BAT energy metabolism through

regulation of theromogenic genes including PPARγ, PGC-1α, Twist1, Dio2, Irx3 etc (Supplementary Table 1).

Of note, alterations between β3-AR mRNA and protein levels are not synchronized within a short term during the desensitization process. For instance, when SVF-derived brown/beige adipocytes were stimulated by a β3-AR agonist for 8 h, we detected changes in β3-AR mRNA (Fig. 1e–g), but not in protein levels (Supplementary Fig. 6b). In addition, β3-AR agonists have been promising pharmaceutical targets for a number of metabolic diseases. For instance, Mirabegron is a β3-AR agonist that has been approved in humans to treat overactive bladder syndrome[68]. However, β3-AR agonists still lack efficacy for combating human obesity. This may, in part, owe to the undesirable side effects of β3-AR desensitization. We submit that further investigation into mechanisms underlying Foxp1-mediated β3-AR regulation will cast new light upon the development of therapeutic strategies for obesity.

## Methods

**Mice**. The Foxp1^{fl/fl} has been used in our previous studies[69]. aP2-Cre (Stock no. 005069 in Jax Lab), Adiponectin-Cre (Stock no. 028020), Thermo Mouse (Stock no. 026690) were obtained from Jax lab. Myf5-Cre mice was kindly provided by Prof. Dahai Zhu in Peking Union Medical College. Transgenic vector containing 5.4 kb promoter of Fabp4 (aP2) gene were kindly provided by Prof. Qiqun Tang in Fudan University. The genetic backgrounds of all knockout mice were C57Bl/6J and the background of transgenic mice were 129S1/Sv. Mice were bred with standard

rodent chow food or HFD. Male mice were used in the experiments unless otherwise indicated. The experiments were not randomized, and the investigators were not blinded to allocation during experiments or outcome assessments. All animal experiments were performed according to the guidelines (SYXK 2011-0112) and received ethical approval from committee of Bio-X Institutes of Shanghai Jiao Tong University.

**Human adipose tissue samples.** Human browning omental adipose tissue for immunohistochemistry was obtained from clinically and pathologically diagnosed patients with pheochromocytoma and sex-, age-, and BMI-matched control subjects. Three samples were analyzed with immunofluorescence or IHC in this study. Primary human stromal vascular fraction (SVF) from subcutaneous WAT (sWAT) for the beige adipocyte induction were obtained from underaged subjects. The adipose tissue pieces were obtained during surgical procedures followed by immediate storage in liquid nitrogen and fixation in formalin. The human study was performed in accordance with relevant guidelines and received ethical approval from the Institutional Review Board of Ruijin Hospital, Shanghai Jiao Tong University School of Medicine. Written informed consent was provided from each participant prior to inclusion in the study.

**Metabolic Study, bioluminescence, and infrared imaging**. Minispec TD-NMR Analysers (Bruker Instruments) were used to evaluate adiposity composition on anesthetized animals. Food intake, energy expenditure, $O_2$ consumption, $CO_2$ production and physical activity were measured by using indirect calorimetry system (Oxymax, Columbus Instruments), installed under a constant environmental temperature (22 °C) and a 12-h light (07:00–19:00 h), 12-h dark cycle (19:00–07:00 h). Mice in each chamber had free access to food and water. The raw data were normalized by body weight and the histograms of day (07:00–19:00 h) and night (19:00–07:00 h) values were the mean value of all points measured during the 12-h period. Bioluminescence analysis for ThermoMouse is performed using Lumina III (Perkin Elmer, IVIS). An infrared camera (T650sc, emissivity of 0.98, FLiR Systems) is placed over top of the anesthetized mouse to acquire static image according to standard protocols.

**Glucose tolerance test (GTT) and insulin tolerance test (ITT)**. For GTT, mice were given i.p. injection of 100 mg/ml D-glucose (2 g/kg body weight) after overnight fasting, and tail blood glucose concentrations were measured by a glucometer (AccuCheck Active, Roche). For ITT, mice were fasted for 4 h before i.p. administration of human insulin (Santa Cruz) (0.75U/kg body weight), and tail blood glucose concentrations were monitored.

**Immunohistochemistry, immunofluorescence, and TEM**. Adipose tissues were fixed in 4% PFA for 24 h at 4 °C, embedded in paraffin or tissue freezing medium (Leica) and sectioned to 8 μm. HE staining was conducted according to standard protocols. For immunofluorescence, heat-induced antigen retrieval with sodium citrate buffer (10 mM sodium citrate, 0.05% Tween 20, pH 6.0) was performed before bone sections were blocked with 10% normal serum containing 1% BSA in TBST (pH 7.6) for 2 h at room temperature, then incubated overnight at 4 °C with primary antibodies to mouse Foxp1 (Millipore, ABE68, 1:100), Ucp1 (Abcam, ab10893, 1:50), Prdm16 (Abcam, ab106410, 1:50). Subsequently, sections were incubated with secondary fluorescent-conjugated or HRP-conjugated antibodies at room temperature for 2 h in the dark. Samples were imaged by the Leica TCS SP5 confocal microscope, Leica DM2500, or Leica 3000B microscope. Transmission electron microscopy (TEM) of white and brown adipose tissue was carried out in accordance with a previous study.

**Cell cultures**. For SVF cell isolation, primary BAT and sWAT were digested with 1 mg ml$^{-1}$ collagenase type I (Sigma) in DMEM (Invitrogen) supplemented with 1% bovine serum albumin for 25 min at 37 °C, followed by density separation. The digestions were quenched with DMEM containing 10% FBS, and filtered through 70-mm filters to remove connective tissues and undigested trunks of tissues. Cells were then centrifuged at $1000 \times g$ for 5 min to separate the SVF cells in the floating layer. The freshly isolated SVF cells were seeded and cultured in growth medium containing DMEM, 20% FBS, 1% penicillin/streptomycin (P/S) at 37 °C with 5% $CO_2$ for 3 days, followed by feeding with fresh medium every 2 days to reach confluence. For brown adipocyte differentiation, the cells were induced with induction medium contains DMEM, 10% FBS, 5 μg ml$^{-1}$ insulin, 0.5 mM iso-butylmethylxanthine (Sigma), 1 μM dexamethasone (Sigma), 50 nM T3 (Sigma) and 5 μM troglitazone (Sigma) for 48 h, and further in growth medium supplemented with insulin, T3 and troglitazone for six days followed by 0.5 mM cyclic AMP (Sigma) treatment for another 4 h. 3T3-L1 (ATCC) and HEK293T (ATCC) were cultured in DMEM with 10% FBS. For 3T3-L1 adipogenic differentiation, cells of 100% confluence were kept in growth medium for 2 days then induced with induction medium for 2 days, after that differentiated in differentiation medium (without T3) for 6 days. For oil red staining, cultured cells were washed with PBS and fixed with 10% formaldehyde for 15 min at room temperature. Then the cells were stained using the Oil red O working solutions (5 g/l in isopropanol) and 4 ml ddH$_2$O for 30 min. After staining, the cells were washed with 60% isopropanol and pictured.

**Adipocyte OCR measurement**. Primary SVF cells from BAT and sWAT were isolated and cultured for 3 days before being plated in XF cell culture microplates (Seahorse Bioscience). SVF cells (10,000 cells) were seeded in each well, and each treatment included cells from three BAT or sWAT replicates. After 6-day differentiation, cultured adipocytes were washed twice and pre-incubated in XF medium (supplemented with 25 mM glucose, 2 mM glutamine and 1 mM pyruvate) for 1–2 h at 37 °C without $CO_2$. The OCR was measured using the XF Extracellular Flux Analyser (Seahorse Biosciences). Oligomycin (2 mM), FCCP (2 mM), and Antimycin A (0.5 mM) were preloaded into cartridges and injected into XF wells in succession. OCR was calculated as a function of time (pmoles per minute per μg protein).

**shRNA lentivirus preparation**. To construct shRNA for Foxp1-shRNA or β3-AR shRNA virus, the forward and reverse oligos for each shRNA were mixed and denatured at 95 °C for 5 min, annealed at 58 °C for 10 min, and then digested by XbaI and BamHI before cloning into lentivirus-based vector pLenti-shRNA-NF1-GFP. The lentivirus were packaged in 293T cells with helper vector pMD2G and psPAX2 by transfecting cells with indicated constructs. After 48-h culture, supernatant was collected by centrifuge at $1000 \times g$ for 5 min, followed by filtered with MILLEX GP. The oligos for shRNA construct were listed in Supplementary Table 2.

**ChIP-seq analysis and Co-IP**. SVF cells were isolated from mouse BAT or sWAT and expanded in passaging cultures for 2 weeks. The ChIP samples were prepared as following[70]. Briefly, 2–5 million cells were cross-linked with 1% formaldehyde (Sigma-Aldrich) for 10 min at room temperature, and then incubated with 125 mM glycine for 5 min to quench the cross-linking reaction. The harvested cells were sonicated on ice to generate DNA fragments of 200–500 bp. The fragmented chromatin fragments were immunoprecipitated with 20 μl of protein A/G magnetic beads (Millipore) coupled with 4 μg of anti-Foxp1 antibody (Millipore, ABE68, 1:200) at 4 °C overnight with rotation. ChIP DNA was treated with protein K (Thermo Fisher) at 55 °C for 6.5 h for decrosslinking. Both ChIP and input DNA libraries were generated using the NEBNext Ultra II DNA Library Prep Kit for Illumina (E7645, NEB). The DNA libraries were amplified for 18 cycles and subjected to sequencing by Illumina sequencer. For ChIP-PCR, purified DNA was quantified using quantitative PCR. The primer sequences are listed in Supplementary Table 2.

For in vitro co-immunoprecipitation (Co-IP), His- or FLAG-tagged proteins were produced in HEK293T transfected by FuGENE with corresponding plasmids. For in vivo Co-IP, SVF cells were isolated and cultured from wild-type mice. Total cell lysates were incubated overnight at 4 °C with antibodies or normal IgG (Santa Cruz) as control. Antibody-antigen complexes were absorbed by Protein A/G PLUS-Agarose (Santa Cruz, sc-2003). After several washes, samples were boiled and analyzed by western blotting.

**qPCR and western blotting**. Total RNA was extracted with Trizol (Invitrogen), reverse transcriptase was used for cDNA generation with the GoScript reverse transcription system (Promega). qPCR was performed with a real time PCR system (ABI 7500) using SYBR Green (Roche). The primer sequences are listed in Supplementary Table 2. For western blotting, cells were lysed and protein samples were incubated with primary antibodies against Foxp1 (Millipore, ABE68, 1:1000), C/ebpβ (Santa Cruz, sc-150, 1:500), UCP1(Abcam, ab10893, 1:1000), PGC-1α (Milipore, ab3242 1:1000), p38 (CST, 9212s, 1:1000), phosphorylated p38 (CST, 9211s, 1:1000), HSL (CST, 4107, 1:1000) and phosphorylated HSL (CST, 4126, 1:1000), β3-AR (myobioscience, MBS253490, 1:1000), His-Tag (MBL, M136-3, 1:2000), FLAG (Agilent, 200471, 1:2000) or β-actin (Selleck, A1016,1:2000) at 4 °C overnight. Proteins were visualized using HRP-conjugated secondary antibody and chemiluminescent HRP substrate (Millipore).

**Luciferase assay**. Luciferase assays were performed in HEK293T or 3T3-L1 cells. The reporter plasmid, *PPARγ-Luc* containing a 2.2 kb fragment of the 5′ flanking region of the PPARγ gene, was obtained from Dr. Hiroshi Takayanagi of the Tokyo Medical and Dental University. *Adrb3-Luc* plasmid containing 4.3 kb of the 5′ flanking region of Adrb3 gene, was constructed by our lab. The expression plasmids of Foxp1 and Cebpβ were constructed into the pcDNA3.0 vector. Cells were transfected using FuGENE HD (Promega) in 24-well plates. The transfection amount of each plasmid was 200 ng, and the total amount of transfected DNA across each transfection was balanced by pcDNA3.0 plasmids when necessary. After 32 h, dual luciferase assay was performed according to the manufacturer's protocols (Promega).

**Statistical analysis**. All data are presented as mean ± SEM. Error bars are SEM. Two-tailed Student's $t$-tests for comparisons between two groups. For all experiments, $P \geq 0.05$ were marked as ns, $P < 0.05$ were considered significant and indicated by *$P < 0.01$ were indicated by **$P < 0.001$ were indicated by ***.

**Reporting summary**. Further information on research design is available in the Nature Research Reporting Summary linked to this article.

## Data availability

All raw western blotting data could be found in the Source Data. Raw ChIP-seq data could be available in NCBI (SRA) database (accessions no: PRJNA547458). All relevant data are available from the authors.

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

## Acknowledgements

This work was supported by research grants from the National Natural Science Foundation of China [91749103, 81421061, 31100624 and 81200586] and grant from the National Major Fundamental Research 973 Program of China [2014CB942902] to X.Z.G., grants from the NIH (R01CA31534), Cancer Prevention Research Institute of Texas (RP120348 and RP120459) and the Marie Betzner Morrow Centennial Endowment to H.O.T.

## Author Contributions

P.L., S.H., S.L., S.X., F.W., W.Z. and R.Z. performed experiments, J.W. and X.G. designed experiments, C.H. and X.Z. helped ChIP-seq analysis, Y.F., N.W., L. H., X.X. and Z.Y. helped preparing samples and instructing experiments, H.T. and J.W. provided human biopsies and mouse lines and revised manuscript, X.G. wrote paper.

## Competing interests

The authors declare no competing interests.

## Additional information

**Peer Review Information** *Nature Communications* thanks the anonymous reviewer(s) for their contribution to the peer review of this work. Peer reviewer reports are available.

