## [Peer Review File · Nature Communications]

Reviewers' comments:

Reviewer #1 (Remarks to the Author):

The authors have identified a novel role for the Foxp1 transcription factor in adipocyte differentiation and thermogenesis. Mechanistically they demonstrate that Foxp1 mediates some of its effects in this process through transcriptional repression of the β 3-adrenergic receptor and interaction with the Prdm16/Cebp β complex. Strengths of the study include the use of multiple in vivo models and also reciprocal modulation of Foxp1 expression levels. The authors propose that further understanding this process may aid the development of therapeutic strategies for obesity and diabetes. This study represents an important contribution to this field of research that could lead to clinical developments with societal impact.

Major comments

- 1) Foxp1 has already been shown to be differentially expressed during adipocyte differentiation, although its enforced expression in 3T3-L1 cells reportedly caused no change in adipocyte differentiation (PMID: 19244248). This citation should be included in the manuscript.
- 2) Figure 1a is described as IHC in the text on page 5 and on the Figure, but the images are immunofluorescence and so this should be corrected
- 3) In Figure 1c it would be helpful to provide a higher magnification of the images as it was hard to determine the localisation of the FOXP1 staining, it appears both nuclei and cytoplasm are stained. It would also be useful to indicate how many PHEO patients were studied and this could be added to the methods.
- 4) In Figure 1i the p38 kinase inhibitor seems to further stimulate Foxp1 expression. This should be mentioned rather than just saying it didn't inhibit expression. Does this have any relevance to the augmented p38 phosphorylation seen in adipose tissues from the Foxp1Ap2 Δ/Δ mice.
- 5) Page 7 – The following sentence should be re-written to indicate that it is an induction of adipocyte differentiation on Foxp1 deletion – the data do not show induction of Foxp1 deletion as stated. 'We observed a marked induction of Foxp1 deletion on brown/beige adipocytes differentiation in SVF progenitors (Fig. 3a, e).'
- 6) Page 11 – I could not understand the following sentence as the text refers to brown adipocytes declining, whereas the figure only shows Adrb3 transcripts. 'As shown in Fig. 6i, the brown adipocytes in Foxp1Ap2 Δ/Δ mice declined more sharply than the down-regulation of β 3-AR mRNA, whereas those of transgenic mice declined more smoothly.'

Minor comments/typos

The manuscript should be re-read carefully for minor typographical errors.

P3 line 9 – membrane not member

P4 line 3 – should be adipocytes not adipocyte

P4 line 4 – β 3-AR should be in italics as refers to gene, or should remove reference to gene when describing an amino acid change

P5 line 2 – should be subpopulations not subpopulation

Figure 1g – Relative is misspelled as Ralative

Page 6 line 5 – it was unclear that cells would be all be treated with CL-316,243 and then in combination with one of the other drugs. Perhaps rephrase as alone with CL-316,243 or in combination with In Figure 1i it would be helpful to use CL as the abbreviation for this treatment alone and not just in the combinations.

Page 7 line 6 – relative not relatively

Page 7 line 8 – Co.1 should be italicised

Page 7 line 9 – remove 'A' and start sentence with Typical

Figure S3e – legend on panel should be temperature not tempreture

Page 9 - reference to Figure S5 specifically mentions WB, however data include QPCR so would be

clearer to just to remove 'western blotting' or to include QPCR.
Page 10 line 9 – profiles not profile, also underlie rather than undermine
Page 12 line 5 – repress not repression
Page 13 line 9, remove 'is'
Page 13 line 18 – understanding not understand
Page 14 last paragraph 'an underaged subjects' should be 'an underaged subject' if only one, or 'underaged subjects' if more than one.
Page 13 end 2nd paragraph should be anesthetized mouse not anesthetic

Reviewer #2 (Remarks to the Author):

The paper by Liu et al. reports the role of Foxp1 in the regulation of brown and beige fat development and whole-body energy expenditure. The authors found that FoxP1 loss leads to increased brown and beige fat differentiation in vivo and vitro. The authors deleted FoxP1 using Myf5-Cre, Ap2-Cre, and Adipo-Cre and found consistent phenotypes. This assures the role of FoxP1 in adipocytes. The authors further showed that FoxP1 is recruited onto the regulatory region of b3-AR gene and represses its transcription by interacting with PRDM16 and CEBPb, the master regulator of brown and beige fat development. This study provides substantial amounts of data to demonstrate that FoxP1 loss leads to an enhanced brown/beige fat development in vivo. This part of the data is overall convincing and significant.

On the other hand, the proposed mechanism, that is desensitization of beta3-adrenergic receptor (b-AR) by FoxP1 is not entirely supported by the provided data. It is partly because FoxP1 deficient cells exhibit higher cell-autonomous ability to differentiate into brown adipocytes or beige adipocytes in cultured conditions where no b-AR agonist exists. If the mechanism of FoxP1 action is through b3-AR sensitivity, would the effect of FoxP1 deficiency emerge only when cells were treated with b3-AR ligands? The authors should clarify this part in detail. Additional comments can be found below that may help the authors to revise this potentially interesting study.

The transcriptional regulation of b3-AR by FoxP1 is compelling, while it remains unclear if the proposed mechanism in Fig.7f applies to all the BAT-genes. The model indicates b3-AR and "etc."- the authors should clarify what the other "etc." genes in the model are. ChIP-seq of FoxP1 would clarify the relevance of this proposed mechanism.

In KO mice, the authors found a relatively modest change in UCP1 expression (~ 3 fold) in the sWAT. On the other hand, the change in the adipose tissue morphology (beige adipocytes) and increased energy expenditure of KO mice were more robust than the degree of UCP1 expression. Does it suggest that FoxP1 loss activate the UCP1-independent thermogenic mechanisms ((calcium cycling or creatine cycling) that have been recently reported? The authors should examine this possibility.

Several isoforms of FoxP1 exists in adipocytes. Do each isoform have a distinct function or binding affinity to the PRDM16-CEBPb complex?

Reply to reviewer's comments by point-to-point.

Reviewer 1:

The authors have identified a novel role for the Foxp1 transcription factor in adipocyte differentiation and thermogenesis. Mechanistically they demonstrate that Foxp1 mediates some of its effects in this process through transcriptional repression of the β 3-adrenergic receptor and interaction with the Prdm16/Cebp β complex. Strengths of the study include the use of multiple in vivo models and also reciprocal modulation of Foxp1 expression levels. The authors propose that further understanding this process may aid the development of therapeutic strategies for obesity and diabetes. This study represents an important contribution to this field of research that could lead to clinical developments with societal impact.

Major comments

1) Foxp1 has already been shown to be differentially expressed during adipocyte differentiation, although its enforced expression in 3T3-L1 cells reportedly caused no change in adipocyte differentiation (PMID: 19244248). This citation should be included in the manuscript.

Answer: Thanks for the reminding of the reviewer. We have cited this manuscript in the second paragraph of Result at Page 5.

“Similarly, qPCR analyses confirmed the down-regulation of *Foxp1* during the 8-hour course of white adipogenic induction in 3T3-L1 cells (Supplemental Figure S1A), which was consistent with previous findings⁵⁴.”

2) Figure 1a is described as IHC in the text on page 5 and on the Figure, but the images are immunofluorescence and so this should be corrected

Answer: We have replaced the description with immunofluorescence on page 5 and all the other places in manuscript.

3) In Figure 1c it would be helpful to provide a higher magnification of the images as it was hard to determine the localisation of the FOXP1 staining, it appears both

nuclei and cytoplasm are stained. It would also be useful to indicate how many PHEO patients were studied and this could be added to the methods.

Answer: We provided the magnified viewer of FOXP1 and Prdm16 immunofluorescence staining for PHEO biopsies sections in supplemental Figure S1b. We performed IHC or immunofluorescence for FOXP1 in three different PHEO samples, which was updated in the methods. From those results, we preferred that Foxp1 mostly located in the nucleus of brown or beige adipocytes.

4) In Figure 1i the p38 kinase inhibitor seems to further stimulate Foxp1 expression. This should be mentioned rather than just saying it didn't inhibit expression. Does this have any relevance to the augmented p38 phosphorylation seen in adipose tissues from the Foxp1Ap2Δ/Δ mice.

Answer: We corrected the description at Page 6 as following: "the induction of Foxp1 transcripts by the β3-AR agonist was markedly reduced by FR180204 and SCH772984, whereas it was exaggerated by SB202190."

We don't know whether the additive effect of p38 kinase inhibitor and CL-316,243 in the induction of Foxp1 expression contribute to the augmented p38 phosphorylation in Foxp1 knockout adipose tissues. The increased p38 phosphorylation in Foxp1-deficient adipocytes may result from the upregulation of Adrb3 and activation of thermogenic program.

5) Page 7 – The following sentence should be re-written to indicate that it is an induction of adipocyte differentiation on Foxp1 deletion – the data do not show induction of Foxp1 deletion as stated. 'We observed a marked induction of Foxp1 deletion on brown/beige adipocytes differentiation in SVF progenitors (Fig. 3a, e).'

Answer: We rewrite the sentence as "We observed an advanced brown/beige adipocyte differentiation from Foxp1-deficient SVF progenitors (Fig. 3a, e)".

6) Page 11 – I could not understand the following sentence as the text refers to brown adipocytes declining, whereas the figure only shows Adrb3 transcripts. 'As shown in Fig. 6i, the brown adipocytes in Foxp1Ap2Δ/Δ mice declined more sharply

than the down-regulation of β 3-AR mRNA, whereas those of transgenic mice declined more smoothly.”

Answer: We rewrite the sentence as “As shown in Fig. 6i, the down-regulation of β 3-AR in brown adipocytes from *Foxp1_{ap2}^{Δ/Δ}* mice declined fast as compared to the wild type controls, whereas those of transgenic mice declined more smoothly.”

Minor comments/typos

The manuscript should be re-read carefully for minor typographical errors.

P3 line 9 – membrane not member

P4 line 3 – should be adipocytes not adipocyte

P4 line 4 – β 3-AR should be in italics as refers to gene, or should remove reference to gene when describing an amino acid change

P5 line 2 – should be subpopulations not subpopulation

Figure 1g – Relative is misspelled as Ralative

Page 6 line 5 – it was unclear that cells would be all be treated with CL-316,243 and then in combination with one of the other drugs. Perhaps rephrase as alone with CL-316,243 or in combination with In Figure 1i it would be helpful to use CL as the abbreviation for this treatment alone and not just in the combinations.

Page 7 line 6 – relative not relatively

Page 7 line 8 – Co.1 should be italicised

Page 7 line 9 – remove ‘A’ and start sentence with Typical

Figure S3e – legend on panel should be temperature not temperture

Page 9 - reference to Figure S5 specifically mentions WB, however data include QPCR so would be clearer to just to remove ‘western blotting’ or to include QPCR.

Page 10 line 9 – profiles not profile, also underlie rather than undermine

Page 12 line 5 – repress not repression

Page 13 line 9, remove ‘is’

Page 13 line 18 – understanding not understand

Page 14 last paragraph ‘an underaged subjects’ should be ‘an underaged subject’ if only one, or ‘underaged subjects’ if more than one.

Page 13 end 2nd paragraph should be anesthetized mouse not anesthetic

Answer: Great thanks for the reviewer's careful examinations in spelling and editions. We went through the manuscript thoroughly and make all the revisions as required. All the revisions are marked as green color in manuscript.

Additional editions was listed as following:

- 1) Page 5, at the last paragraph, we added a sentence "These observations suggest that *Foxp1* expression in adipose adipocytes was dynamic, and can be induced in an inverse expression pattern with that of β -AR by adrenergic signaling."
- 2) All the spelling of "Cebp β " was changed into "C/ebp β ", the Prdm16/ Cebp β complex was changed into "Prdm16-C/ebp β ".
- 3) The spelling of "Ap2-Cre" was changed into "aP2-Cre", which was more frequently used in publications.

Reviewer 2:

The paper by Liu et al. reports the role of Foxp1 in the regulation of brown and beige fat development and whole-body energy expenditure. The authors found that FoxP1 loss leads to increased brown and beige fat differentiation in vivo and vitro. The authors deleted FoxP1 using Myf5-Cre, Ap2-Cre, and Adipo-Cre and found consistent phenotypes. This assures the role of FoxP1 in adipocytes. The authors further showed that FoxP1 is recruited onto the regulatory region of b3-AR gene and represses its transcription by interacting with PRDM16 and CEBPb, the master regulator of brown and beige fat development. This study provides substantial amounts of data to demonstrate that FoxP1 loss leads to an enhanced brown/beige fat development in vivo. This part of the data is overall convincing and significant.

On the other hand, the proposed mechanism, that is desensitization of beta3-adrenergic receptor (b-AR) by FoxP1 is not entirely supported by the provided data. It is partly because FoxP1 deficient cells exhibit higher cell-autonomous ability to differentiate into brown adipocytes or beige adipocytes in cultured conditions where no b-AR agonist exists. If the mechanism of FoxP1 action is through b3-AR sensitivity, would the effect of FoxP1 deficiency emerge only when cells were treated with b3-AR ligands? The authors should clarify this part in detail. Additional comments can be found below that may help the authors to revise this potentially interesting study.

Answer: The reviewer has put up the critical point of the mechanisms of Foxp1 in regulating BAT activation and thermogenesis. We think that Foxp1 may have multiple targets to exert its effect on thermogenic program. β 3-AR is only one of the Foxp1 targets. ChIP-seq analysis revealed other BAT-selective gene targets including *PGC-1 α* , *Dio2*, etc. We clarified the statement in the Discussion part at Page 13 as following:

“Of note, it has to be kept in mind that the β 3-AR pathway only partially account for the function of Foxp1 in BAT activation and thermogenesis, given that *Foxp1*-deficient cells exhibit higher cell-autonomous ability to differentiate into brown adipocytes or beige adipocytes in cultured conditions without β 3-AR agonist stimulation. ChIP-seq analysis also suggested that Foxp1 may control BAT energy metabolism through regulation of thermogenic genes including *PPAR γ* , *PGC-1 α* , *Twist1*, *Dio2*, *Irx3* etc (Supplemental Table S1).”

The transcriptional regulation of β_3 -AR by FoxP1 is compelling, while it remains unclear if the proposed mechanism in Fig.7f applies to all the BAT-genes. The model indicates β_3 -AR and "etc."- the authors should clarify what the other "etc." genes in the model are. ChIP-seq of FoxP1 would clarify the relevance of this proposed mechanism.

Answer: ChIP-seq of FoxP1 in SVF-BAT cells revealed that Foxp1 may regulate the expression of an array of BAT-selective genes, such as *PPAR γ* , *PGC-1 α* , *Twist1*, *Dio2*, *Irx3*, etc. Therefore, we clarified the other "etc." genes in the revised model.

In KO mice, the authors found a relatively modest change in UCP1 expression (~ 3 fold) in the sWAT. On the other hand, the change in the adipose tissue morphology (beige adipocytes) and increased energy expenditure of KO mice were more robust than the degree of UCP1 expression. Does it suggest that FoxP1 loss activate the UCP1-independent thermogenic mechanisms ((calcium cycling or creatine cycling) that have been recently reported? The authors should examine this possibility.

Answer: I agreed with the reviewer's point that the alteration of Ucp1 expression only partially accounted for the robust phenotype of thermogenesis in *Foxp1* KO mice. However, as showed in the following figure, we examined the calcium or creating cycling-related thermogenesis in *Foxp1*-KO adipose tissues by qPCR with a number of marker genes (Calcium-dependent: *Ryr2*, *Serca1*, *Serca2b*, *Serca3*; Creatine-dependent: *Ckmt1*, *Gamt1*, *Gatm*, *Slc6a8*). Only *Gatm* expression was slightly increased in *Foxp1* mutant white adipose tissues, but not in BAT. Therefore, the contributions of calcium or creatine-dependent pathway to the defective thermogenesis in *Foxp1* KO mice remains relatively limited.

Ca²⁺-dependent Markers in BAT

Ca²⁺-dependent Markers in sWAT

Creatine-related markers in BAT

Creatine-related markers in sWAT

Several isoforms of FoxP1 exists in adipocytes. Do each isoform have a distinct function or binding affinity to the PRDM16-CEBPb complex?

Answer: There are 4 Foxp1 isoforms of A, B, C, D. The expressing vector of Foxp1 we used for Co-IP and luciferase reporter harbored Foxp1 isoform2 from NCBI database (NP_001184250.1). We performed luciferase reporter with *Adrb3*-Luc and *Cebpβ*, in combination with Foxp1, Foxp1A, B, C, D, respectively. The assay indicated that Foxp1B may be the major isoform that controls BAT activation and thermogenesis. We added the data into the Discussion part of manuscript as Supplemental Fig. S5d.

REVIEWERS' COMMENTS:

Reviewer #1 (Remarks to the Author):

The revisions to this manuscript address the concerns raised in my previous review.

Reviewer #2 (Remarks to the Author):

The authors provide new data, such as ChIP-seq, that support the authors' conclusion. I have two minor comments that the authors should address.

1. Since Adiponectin-Cre gives consistent data, I would suggest replacing the Ap2-Cre mice data with the Adipo-Cre data. Adipo-Cre is more selective to adipocytes than Ap2-Cre.
2. A recent paper shows that beige fat is heterogeneous and contains a glycolytic form of beige fat (g-beige) whose development is regulated independently of b3-AR signaling. Since Foxp1 controls the b3-AR pathway, the authors should discuss or examine if g-beige fat (its molecular markers) is regulated in the WAT of Foxp1 KO and overexpressed mouse models.

point-by-point response :

Reviewer #2 (Remarks to the Author):

The authors provide new data, such as ChIP-seq, that support the authors' conclusion. I have two minor comments that the authors should address.

1. Since Adiponectin-Cre gives consistent data, I would suggest replacing the Ap2-Cre mice data with the Adipo-Cre data. Adipo-Cre is more selective to adipocytes than Ap2-Cre.

Answer: We have placed all the Adipo-Cre data in main text, whereas all aP2-Cre data was put in supplemental data.

2. A recent paper shows that beige fat is heterogeneous and contains a glycolytic form of beige fat (g-beige) whose development is regulated independently of b3-AR signaling. Since Foxp1 controls the b3-AR pathway, the authors should discuss or examine if g-beige fat (its molecular markers) is regulated in the WAT of Foxp1 KO and overexpressed mouse models.

Answer: We have performed qPCR for WAT of Foxp1 KO mice with glycolytic beige adipocyte markers. Similar to the expression profiles of other brown and beige adipocyte markers, we also observed elevated expression of MyoD, Myh1, GABP α in WAT of Foxp1_{ap2}^{Δ/Δ} mice (Supplemental Figure S6a). Yet we did not observed obviously decreased expression of these genes in WAT from aP2-Foxp1 transgenic mice, which may require β -blocker treatment and prolonged cold challenge. It suggests that Foxp1 also regulates the β -AR-independent beige adipocyte activation. We discussed this result in the updated version of manuscript.